# CD44v6 Defines a New Population of Circulating Tumor Cells Not Expressing EpCAM

**DOI:** 10.3390/cancers13194966

**Published:** 2021-10-02

**Authors:** Guillaume Belthier, Zeinab Homayed, Fanny Grillet, Christophe Duperray, Julie Vendrell, Ilona Krol, Sophie Bravo, Jean-Christophe Boyer, Olivia Villeronce, Jihane Vitre-Boubaker, Diana Heaug-Wane, Françoise Macari-Fine, Jai Smith, Matthieu Merlot, Gérald Lossaint, Thibault Mazard, Fabienne Portales, Jérôme Solassol, Marc Ychou, Nicola Aceto, Emilie Mamessier, François Bertucci, Jean Marc Pascussi, Emmanuelle Samalin, Frédéric Hollande, Julie Pannequin

**Affiliations:** 1Institute of Functional Genomics (IGF), UMR5203 CNRS, U1191 INSERM and UM, 34094 Montpellier, France; guillaume.belthier@free.fr (G.B.); zeinab.homayed@igf.cnrs.fr (Z.H.); fanny.grillet@gmail.com (F.G.); Olivia.Villeronce@igf.cnrs.fr (O.V.); jihane.vitre@gmail.com (J.V.-B.); diana.heaug@gmail.com (D.H.-W.); Francoise.Macari@igf.cnrs.fr (F.M.-F.); Jean-Marc.Pascussi@igf.cnrs.fr (J.M.P.); emmanuelle.samalin@icm.unicancer.fr (E.S.); 2Cytometry IRMB, Montpellier Rio Imaging, 34090 Montpellier, France; christophe.duperray@inserm.fr; 3Department of Pathology and Onco-Biology, CHU Montpellier, 34295 Montpellier, France; j-vendrell@chu-montpellier.fr (J.V.); j-solassol@chu-montpellier.fr (J.S.); 4Department of Biology, Institute of Molecular Health Sciences, ETH Zurich, 8093 Zurich, Switzerland; ilona.krol@biol.ethz.ch (I.K.); nicola.aceto@biol.ethz.ch (N.A.); 5Laboratoire de Biochimie, CHU Carémeau, 30900 Nîmes, France; sophie.bravo@chu-nimes.fr (S.B.); jean.christophe.boyer@chu-nimes.fr (J.-C.B.); 6Department of Clinical Pathology, Victorian Comprehensive Cancer Centre, The University of Melbourne, Melbourne, VIC 3010, Australia; jai.smith@unimelb.edu.au (J.S.); frederic.hollande@unimelb.edu.au (F.H.); 7University of Melbourne Centre for Cancer Research, Victorian Comprehensive Cancer Centre, Melbourne, VIC 3010, Australia; 8Medical Oncology Department, Institut du Cancer de Montpellier (ICM), University Montpellier, 34298 Montpellier, France; Matthieu.Merlot@icm.unicancer.fr (M.M.); Gerald.Lossaint@icm.unicancer.fr (G.L.); Thibault.Mazard@icm.unicancer.fr (T.M.); Fabienne.portales@icm.unicancer.fr (F.P.); Marc.Ychou@icm.unicancer.fr (M.Y.); 9Montpellier Research Cancer Institute (IRCM), INSERM U1194, University of Montpellier, 34298 Montpellier, France; 10Predictive Oncology Laboratory, Cancer Research Center of Marseille (CRCM), Inserm U1068, CNRS UMR7258, Institut Paoli-Calmettes, Aix Marseille Université, 13009 Marseille, France; emilie.mamessier@inserm.fr (E.M.); bertuccif@ipc.unicancer.fr (F.B.)

**Keywords:** circulating tumor cells, colorectal cancer, breast cancer, biomarker, patient blood samples, EMT

## Abstract

**Simple Summary:**

In the present work, we describe (for the first time) the use of the transmembrane protein, CD44v6, to detect CTCs from blood samples of several patients with colorectal or breast cancer. We used CD44v6 antibodies to demonstrate that live CTCs can be specifically purified from CRC patient blood samples via magnetic bead- or FACS-based isolation techniques. Finally, we demonstrated that CD44v6-positive CTCs rarely expressed EpCam, which is currently the gold standard to enumerate CTCs, suggesting the need to use a combination of markers for a more comprehensive view of CTC heterogeneity.

**Abstract:**

Circulating tumor cells (CTCs) are promising diagnostic and prognostic tools for clinical use. In several cancers, including colorectal and breast, the CTC load has been associated with a therapeutic response as well as progression-free and overall survival. However, counting and isolating CTCs remains sub-optimal because they are currently largely identified by epithelial markers such as EpCAM. New, complementary CTC surface markers are therefore urgently needed. We previously demonstrated that a splice variant of CD44, CD44 variable alternative exon 6 (CD44v6), is highly and specifically expressed by CTC cell lines derived from blood samples in colorectal cancer (CRC) patients. Two different approaches—immune detection coupled with magnetic beads and fluorescence-activated cell sorting—were optimized to purify CTCs from patient blood samples based on high expressions of CD44v6. We revealed the potential of the CD44v6 as a complementary marker to EpCAM to detect and purify CTCs in colorectal cancer blood samples. Furthermore, this marker is not restricted to colorectal cancer since CD44v6 is also expressed on CTCs from breast cancer patients. Overall, these results strongly suggest that CD44v6 could be useful to enumerate and purify CTCs from cancers of different origins, paving the way to more efficacious combined markers that encompass CTC heterogeneity.

## 1. Introduction

According to recent data, 9.6 million people die each year from cancer, with the vast majority of these deaths being due to metastasis to distant organs [1]. The development of metastases is a challenging, multi-step process for tumor cells, starting with their ability to disseminate from the primary tumor and concluding with the colonization of distant organs and the formation of metastatic lesions. To reach metastatic sites, tumor cells usually transit through the bloodstream, where they are called circulating tumor cells (CTCs). CTCs have been proposed, for several years, as promising biomarkers. Their detection and quantification were proven clinically useful in diagnostic and prognostic settings as well as to monitor treatment responses and predict tumor recurrence [2,3].

Several approaches have been used to isolate CTCs from cancer patient blood samples, such as the ISET technique, which is based on larger sizes of CTCs compared to hematopoietic cells [4]. Microfluidic CTC capture platforms have also been engineered to sort rare CTCs from whole blood, such as the iChip, which enables the isolation of CTCs by strategies that are either dependent or independent of tumor markers and thus theoretically applicable to all cancers [5]. Similarly, the Angle plc Company (UK) has developed an epitope-independent technology that enables the isolation of CTCs thanks to their different sizes and deformability [6]. The latter technology has allowed the key discovery of both CTC clusters and CTC-neutrophil clusters [7,8]. Nevertheless, the CellSearch platform remains the clinical gold-standard, and it is the only FDA-approved method used to count CTC in blood samples as a prognostic clinical marker, notably for metastatic colorectal (CRC), prostate, and breast cancers [9,10,11]. However, the CellSearch approach necessitates the fixation of cells, thus precluding their subsequent maintenance in culture. In addition, the platform is based on the detection of epithelial markers, such as EpCAM and cytokeratins, combined with the lack of CD45 expression in nucleated, circulating cells. However, the latter technique might not be exhaustive, and consequently, some CTCs could be ignored. Indeed, the epithelial–mesenchymal transition (EMT) has been implicated in the dissemination process [12,13,14]. EMT naturally leads to the loss of epithelial marker expressions, including EpCAM. Clearly, then, CTCs are heterogeneous and understanding them better would be precious to combatting cancer mortality.

CTCs have been enriched and sometimes cultured from several types of cancers [15,16,17,18]. In CRC, we were the first to establish and characterize several CTC lines derived from patient blood samples [19]. We notably demonstrated that CTCs highly and specifically expressed an alternative epitope of the CD44 protein that came from the splicing of alternative variable exon 6 (v6). This CD44v6 isoform had been previously described as a marker of cancer cells driving metastasis [20]. In addition, CD44v6 has been described, in pancreatic cancer, as one of the markers expressed on migrating cancer-initiating cells [21]. CD44v6 is also notably expressed in gastric cancer [22]. For several solid cancers, the CD44v6-positive cell percentage within the tumor inversely correlates with overall survival [23,24,25,26]. An important meta-analysis confirmed these data and associated CD44v6 expression in the primary tumor with distant metastasis [27]. Loss and gain of function experiments demonstrated, a long time ago, that the v6 isoform is key to the ability to develop metastasis [28].

As CD44 is a transmembrane protein, we investigated the possibility of using CD44v6 as bait to purify CTCs from patient blood samples. We discovered that CD44v6 was a promising marker to isolate some CTCs from CRC patient blood samples, and CD44v6-expressing CTCs were a distinct population from EpCAM-positive CTCs. Finally, CD44v6 was highly expressed in CTC from patients with breast cancer and thus may have a broader clinical value.

## 2. Results

CD44v6 and EpCAM are mutually exclusively expressed in viable non-hematopoietic cells from CRC patient blood samples.

To search for clinically relevant circulating tumor cell (CTC) epitopes, we first co-stained the CD45-negative (CD45^−^) fraction of CRC blood samples from 13 patients with our potential CTC marker, CD44v6, and the known CTC marker EpCAM. Strikingly, most CD45^−^ cells were distinctly mono-labelled with either CD44v6 or EpCAM but not both (Figure 1). We, therefore, wondered if these CD44v6-positive cells could be different CTCs that were missed by EpCAM-based detection. To ascertain if CD44v6 could be used as a cell surface marker to isolate CTCs from blood samples, we thus used two techniques: magnetic beads and cytometry.

Magnetic bead-based isolation with a CD44v6 antibody efficiently discriminates tumor cells from hematopoietic cells. Our magnetic bead approach is summarized in Figure 2A and detailed in the Materials and Methods section. Briefly, we isolated the rare CD44v6-positive cells by labelling the cells with an Allophycocyanin (APC) pre-coupled antibody directed against the CD44v6 membrane protein, and then, we incubated the cells with beads, coupled with an anti-APC antibody. The magnetic column allows us to retain, and thus enrich, CD44v6-positive cells. To approximate the ratio of CTCs to hematopoietic cells in the blood of CRC patients, we spiked 100 CTCs—from the previously established CRC patient blood cell line (CTC45)—into 6 mL of the blood of healthy individuals, and red blood cell lysis was performed to remove erythrocytes. Following elution from magnetic columns, CD44v6 staining was performed, and its expression was assessed by cytometry. The recovery ratio was about two-thirds of the 100 spiked-in cells (63.3 ± 3.7%, *n* = 3).

Next, we extracted CD44v6-positive blood samples from 20 colorectal cancer patients. For all samples, this approach succeeded in isolating live cells that could grow and multiply for several days before dying (live cells are shown in Figure 2B). For one of these samples, we even managed to establish a novel CTC line. The BRAFV600E mutation that is frequent in CRC was detected in this novel cell line through pyrosequencing (Figure 2C). In addition, a tumor was formed following injection of these cells into the flank of nude mice, confirming their tumoral origin (Figure 2D). All together, these results demonstrate that CTCs can be purified from CRC patient blood samples with magnetic beads based on the expression of CD44v6.

Cytometry-based isolation following CD44v6 staining efficiently discriminated tumor cells from hematopoietic cells.

The second strategy was to use cytometry to isolate CTCs. First, for this strategy, to facilitate tracking during the entire process, we transduced a GFP-expressing vector into the CTC45 line before spiking them into healthy erythrocyte-cleared blood, as described above. We then co-stained the mixture with an APC-conjugated CD44v6 antibody to detect CTCs by FACS and a Phycoerythrin (PE)-conjugated CD45 antibody to exclude white blood cells (Figure 3A). The Near-IR Dead Cell Stain Kit was also used to exclude dead cells. A CD44V6^high^ gate was delimited by passing the erythrocyte-free blood through the cytometer without any prior staining. Spiked CTC45 cells expressing GFP were successfully purified based on the expression of CD44v6 from the blood samples of healthy patients, confirming their origin and purity (green spheres were obtained in the suspension culture; Figure 3B). Then, CD44v6 expression was assessed by flow cytometry following cell sorting, and the same recovery rate of up to two-thirds, as found for bead purification above, was obtained (65.7 ± 1.19 %, *n* = 3; Appendix A).

Protocols for the two approaches are detailed in the Material and Methods section. For the ensuing work, we preferred the cytometry strategy for cell isolation because of the noticeable contamination of CD45-positive cells with magnetic beads that rendered CTC fine enumeration impossible.

CD44v6^high^/CD45^−^ CTCs can be purified from CRC patient blood samples by cytometry.

To test if CD44v6 allows the isolation of CTCs, the number of CD45^negative^/CD44V6^high^ (CD45^−^/CD44V6^high^) cells was then quantified by flow cytometry analysis in CRC patient blood samples. This experiment was performed on blood samples from 13 treatment-naïve patients with late-stage CRC bearing various mutational profiles and clinical parameters (Table 1). Potential CTCs (CD45^−^/CD44V6^high^) could be detected in each sample, and they were significantly more frequent in CRC patient blood samples than in healthy ones (*p* = 0.0001; Figure 4A). Residual CD44v6 expression is likely due to its rare expression on normal leukocytes [29].

Next, we further analyzed the mutational profiles of sorted CD45^−^/CD44V6^high^ cells from three patient blood samples. First, knowing that one of the patients’ no. 124 primary tumor had a KRAS mutation detected with next-generation sequencing on DNA from 62 CD45^−^/CD44V6^high^ cells, we performed droplet digital PCR, as described in the Appendix A (Figure 4B), from this patient blood sample to specifically search for this mutation (exon 2, G12V). This highly sensitive technique enabled us to detect two copies of mutated KRAS G12V (Figure 4C), further validating the colorectal tumor origin of the isolated cells. Importantly, CTCs that were wild type for this gene were also found, suggesting that CD44v6 based isolation could give us an idea of mutational tumor heterogeneity for each patient. Similarly, from two CRC patient blood samples, no. 178 and no. 99 (Figure 4D and Appendix A), 329 and 97 CD45^−^/CD44V6^high^ cells were sorted. DNA was then extracted to annotate mutations by next-generation sequencing, as described in the Appendix A, and CRC mutations in FGFR1, ERBB4, STK11, and PTEN were detected (Table 2 and Figure 4E). This latter result further confirmed that the purified cells from the blood had a tumoral origin.

CD44v6 is also expressed on circulating breast cancer cells.

Next, we investigated the potential organ specificity of this newly identified CTC marker. We used flow cytometry to assess the expression of CD44v6 on two breast cancer patient-derived CTC lines established by our collaborators (Br16 and Brx50) [30]. The majority of the cells on both lines expressed CD44v6, indicating that this marker is not only expressed by colorectal CTCs but also strongly expressed in CTC lines from a cancer of a different origin (Figure 5A). Finally, to exclude the possibility that high CD44v6 expression on breast CTCs was the consequence of a culture bias, blood samples from three breast cancer patients were tested for the presence of CD45^−^/CD44V6^high^ cells. Figure 5B shows that CTCs (CD45^−^/CD44V6^high^ cells) were found in the blood of three patients, and their number was significantly higher than in healthy subjects (*p* = 0.0045; Appendix A).

Overall, the data presented here strongly suggest that CD44v6 is a robust transmembrane CTC candidate marker in colorectal, breast and potentially other cancers.

## 3. Discussion

In the present work, we detected CD45^−^/CD44V6^high^ cells in all colorectal cancer (CRC) patient blood samples, irrespective of clinical data (Table 1), suggesting that this antigen could be a precious tool for providing a better overview of tumor heterogeneity.

Circulating tumor cells (CTCs) are widely counted in clinical blood samples, and the research focus has now moved toward better isolating and characterizing these CTCs to elucidate mechanisms of tumor dissemination and metastases. These studies are leaning toward the enticing prospect of isolating CTCs and performing drug screening to thereafter select the most adapted therapy and personalized patient treatment. This concept was most notably suggested by Yu and colleagues in 2014 in breast cancer [18]. A major obstacle to reach these goals is the scarcity of CTCs in circulation, as very few of them can usually be detected amongst millions of hematopoietic cells. It is currently difficult to efficiently purify CTCs from patient blood samples due to the paucity of reliable, selective markers.

By depleting blood cells from patient samples and culturing remaining cells in suspension, we were the first to isolate and culture several CTC lines from CRC patient blood samples [19]. During this work, we used flow cytometry to demonstrate that CD44v6 was highly expressed in almost all cells of cultured CTC in the lab. In the intestine, CD44 is one of the multiple target genes of the Tcf-4/β-catenin pathway, constitutively activated in almost 80% of CRCs, but specific isoforms of this gene could be also detected in CRCs. In 2014, Todaro and his team demonstrated that CD44v6 marked cancer cells that drive colon cancer metastasis [20]. Similarly, CD44v6 was described to be related to the poor outcome of patients with colorectal cancer via an upregulation of the mesenchymal phenotype [26].

From these previous studies, the idea emerged that CD44v6 could be used as a biomarker to specifically isolate CTCs from the blood of patients with CRC. Indeed, Nicolazzo and colleagues recently proposed that CD44v6 could be used as a treatment-failure marker in metastatic CRC patient blood samples. However, in the latter publication, they did not isolate CD44v6-positive cells to prove their tumoral origin [31].

Importantly, the CD44 gene is also expressed in normal human peripheral blood [29], which, in the present work, strengthens the relevance of enumerating/isolating cells expressing the CD44V6 isoform in the CD45^negative^ cell subpopulation.

In addition, a study in 2000 highlighted the limitations of CD44v6 amplification for the detection of tumor cells in the blood of CRC patients [32], but more fundamentally, this work only considered CD44v6 expression at the mRNA level, which could considerably differ from protein expression.

In the present work, we demonstrate that CD44v6 can be successfully used to detect CTCs from the blood of not only colorectal cancer but also breast cancer patients, suggesting that this marker could be expressed by CTCs from cancers of multiple origins. Results could thus be extended to CTC in other cancers and potentially become a pan-transmembrane CTC marker.

It is very interesting that EpCAM and CD44v6 mark distinct CTC subpopulations, and it will thus be very interesting to study the differential properties of these two subpopulations. Clearly, the v6 population is likely to be more aggressive and more stem-like according to our previous data [19]. To move personalized medicine forward, it is compulsory to have access to the most exhaustive population of CTCs. Here, we demonstrate that CTCs co-expressing EpCam and CD44v6 are extremely rare, confirming the importance of combining markers to encompass CTC heterogeneity.

Other specific, common markers for CTCs have been suggested in the literature in the last few years. In colorectal cancer, Plastin-3 has been proposed but has not yet been validated as a robust marker [33]. EGFR has also been suggested as a promising candidate that is expressed on CTCs of 15% of metastatic CRC patients. However, this marker was found in 10% of healthy individuals, which undermines its specificity [34]. More convincingly, the carcinoembryonic antigen (CEA) is expressed by CTCs in 66% of patients with metastatic CRC, although CEA is detected in the blood of almost 20% of healthy patients [35]. Recently, VAR2CSA, another transmembrane marker, has been proposed to isolate CTC [36].

Overall, our work is the first demonstration that so far ignored CTCs from the blood of patients with various cancers can be purified by exploiting their high expressions of CD44v6. Further work is now needed to functionally characterize this novel CTC subpopulation.

## 4. Material and Methods

Blood samples: Blood samples from healthy donors were obtained from the Etablissement Français du Sang (EFS) blood bank under agreement no. 21PLER2016-0013. Blood samples from metastatic colorectal patients were obtained from the Montpellier Cancer Institute (Institut du Cancer de Montpellier; ICM) with patients’ signed consent under ethics agreement number 2016-A00080-51. Blood samples from metastatic breast cancer patients were obtained from the Paoli-Calmettes Institute (Marseille, France) with patients’ signed consent within the PERMED-01 trial (NCT02342158).

Red Blood Cell Lysis: One volume of Red Blood Cell Lysis Solution (10X) (130-094-183, Miltenyi Biotec, Bergisch Gladbach, Germany) was diluted in double-distilled water (ddH2O). One volume (6 mL) of blood was then diluted 10-fold in Red Blood Cell Lysis Solution 10-fold and vortexed for 5 s. After a 10-min incubation period at room temperature, the sample was centrifuged at 300× *g* for 10 min (this speed and timing of centrifugation are the same for the following experiments). The supernatant was finally removed, and the cell pellet was suspended with the blocking buffer.

Blocking Buffer: This buffer contains the auto MACS rinsing solution (130-091-222, Miltenyi Biotec) with 1% of BSA (130-091-376, Miltenyi Biotec).

Magnetic bead-based enrichment: First, the cells were counted and centrifuged. Then, the pellet was suspended to reach the concentration of 10^7^ cells in 70 µL of blocking buffer. Then, 20 µL of Human FcR Blocking Reagent (130-059-901, Miltenyi Biotec) was added and incubated on ice for 10 min. Cells were washed with the blocking buffer and then centrifuged to perform the antibody staining. A total of 10 µL of Allophycocyanin (APC)-conjugated antibody (130-111-425, Miltenyi Biotec), for up to 10^7^ cells, was added for a final volume of 100 µL. Samples were incubated for 10 min on ice and then washed twice with 1 mL of blocking buffer. After another centrifugation step, the pellet was suspended at 10^7^ cells in 80 µL of blocking buffer. Then, 20 µL of anti-APC microbeads (130-090-855, Miltenyi Biotec) were added and incubated on ice for 15 min and then washed twice with 1 mL of blocking buffer. After another centrifugation step at 300 g, the cell pellet was suspended in 500 µL of blocking buffer for up to 10^8^ cells. Cells were placed on LS Columns (130-042-401, Miltenyi Biotec) that had been previously introduced into the QuadroMACS Separator (130-090-976, Miltenyi Biotec) and equilibrated with 500 µL of blocking buffer. The LS Columns were then washed 3 times and removed from the separator to be placed on a new collection tube. Cells of interest were eluted with 2 mL of PBS.

Flow cytometry and sorting: Following Red Blood Cell Lysis, cells were first counted and centrifuged. The pellet was suspended to reach the concentration of 10^7^ cells in 80 µL of blocking buffer. Then, 20 µL of Human FcR Blocking Reagent (130-059-901, Miltenyi Biotec) was added and incubated on ice for 10 min. Cells were washed with blocking buffer and then centrifuged to proceed to antibody staining.

There was a total of 10 µL of APC-conjugated CD44v6 antibodies (130-111-425, Miltenyi Biotec) and Vio Bright B515-conjugated EpCAM antibodies (130-111-007). The dilution was 1:50 for up to 10^6^ cells in a final volume of 100 µL of blocking buffer.

The dilution for PE-conjugated CD45 antibodies (FAB1430P-100, R&D SYSTEMS) was 10 μL for 10^6^ cells. The Live/Dead Fixable Near-IR Dead Cell Stain Kit (L34976, Invitrogen, Walthman, MA, USA) dilution was 1 μL for 10^6^ cells. Samples were incubated for 15 min on ice and then washed twice with 1 mL of blocking buffer. The 10^6^ cells were suspended with 200 µL of blocking buffer. Cells of interest were sorted using FACS ARIA IIIU from BECTON DICKINSON with FACSDiva Software into a microtube containing 3 μL of alkaline lysis buffer (200 mM KOH, 50 mM DTT). Cells were lysed by incubation at 65 °C for 10 min, then stored at −20 °C until next-generation sequencing or digital droplet PCR was performed.

For CD44v6 staining on breast CTC lines, patient-derived cell lines were used as negative and positive controls. Breast CTC lines were stained with APC-conjugated CD44v6 antibodies (130-111-425, Miltenyi Biotec) and then analyzed using BD LSR Fortessa with BD FACSDiva 8.0 Software. CD44v6 staining was compared to the unstained condition. Dead cells were excluded with DAPI staining. All cell lines were cultured in the same conditions and dissociated as described below.

Cell culture and sphere dissociation: Cells were seeded in M12 medium (1 mL/well) in ultralow attachment flasks (Corning) that enabled sphere formation. M12 medium contained advanced DMEM-F12 (Gibco), 2 mmol/L of L-glutamine, 100 Unit/mL of penicillin and streptomycin, N2 supplement (Gibco), 20 ng/mL of epidermal growth factor (R&D) and 10 ng/mL of fibroblast growth factor-basic (R&D). Spheres were centrifuged at 300 g for 5 min, and the supernatant was removed. Pellets were suspended with an equal volume of Accumax (A7089-100ML Sigma-Aldrich, St Louis, MO, USA). Spheres were then gently mixed and incubated for 30 to 45 min according to the sphere size at 37 °C. For Accumax inactivation, 3 mL of blocking buffer was added and finally passed through a 40 μm Falcon Cell Strainer before counting and staining.

Statistics. All graphs and statistical analyses were performed with GraphPad Prism Software. Statistics for CD44v6 expression were expressed as the mean (±standard deviation). The number of living CD45^−^/CD44v6^high^ cells in the different groups was compared using the nonparametric, unpaired Mann–Whitney U test (2-tailed, confidence interval of 95%).

## Figures and Tables

**Figure 1 cancers-13-04966-f001:**
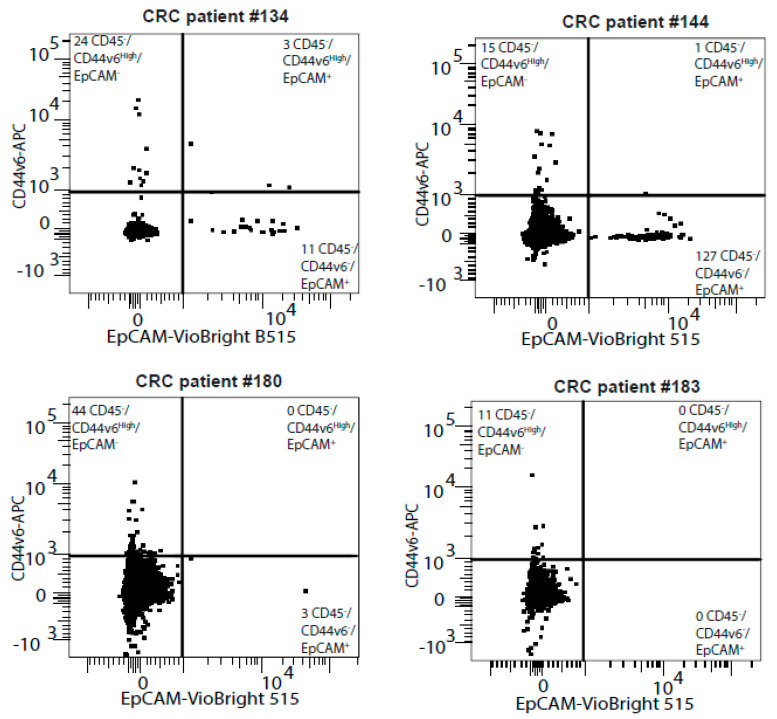
CD44v6 and EpCAM are mutually exclusively expressed in CRC patient blood samples. The four representative FACS plots quantify CD44v6 and EpCAM expression in the CD45-negative cell population from CRC patient blood samples.

**Figure 2 cancers-13-04966-f002:**
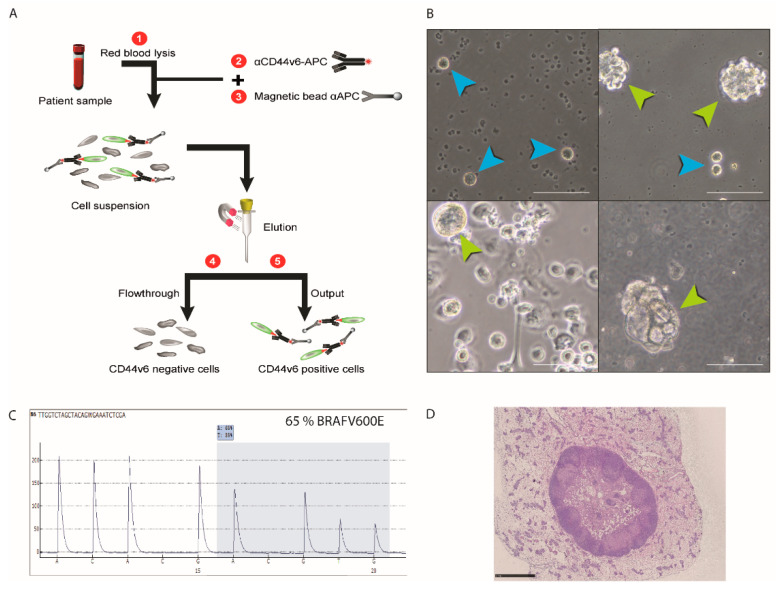
Magnetic bead-based isolation using an antibody directed against CD44v6 efficiently discriminated tumor cells from hematopoietic cells. (**A**) Schematic flowchart summarizing the strategy used to purify CTCs from blood using magnetic beads. Detailed protocol is described in the Materials and Methods section. APC stands for “Allophycocyanin”. (**B**) Photos of live cultured CTCs from four CRC patient blood samples following labelling with antibodies directed against CD44v6 and purification with magnetic beads. Blue arrowheads show single cells, and green arrow heads show spheres initiated from proliferative single cells. Scale bars represent 50 µm. (**C**) Detection of the BRAF V600E mutation in the newly established CTC line by pyrosequencing is described in the Appendix A. In the mutated samples, “A” at position 600 replaces “T”. (**D**) Hematoxylin/eosin staining on a subcutaneous tumor obtained after the novel CTC line was injected in the flank of a nude mouse.

**Figure 3 cancers-13-04966-f003:**
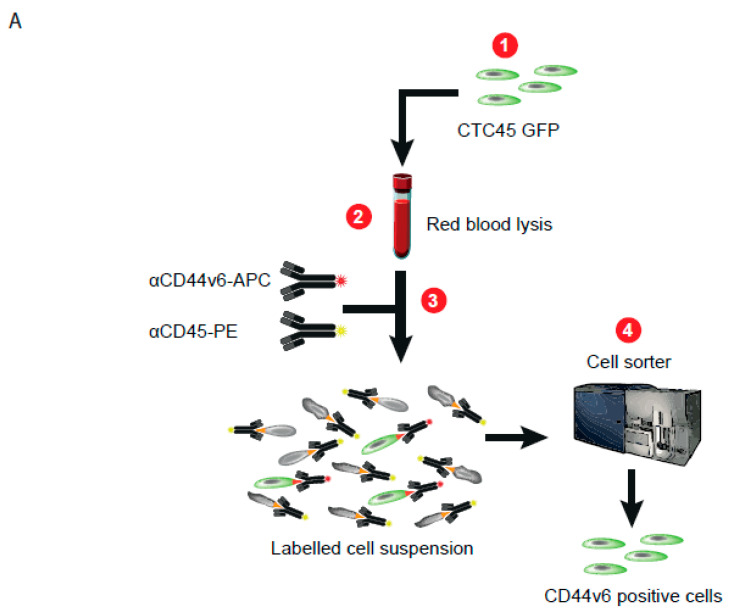
Cytometry-based isolation following CD44v6 staining efficiently discriminated tumor cells from hematopoietic cells. (**A**) Flowchart summarizing the strategy used to purify CTC in patient blood samples through fluorescent-activating cell sorting (FACS). Detailed protocol is described in the Materials and Methods section. (**B**). Photos of GFP-positive CTCs cultured after cell sorting following staining with an antibody directed against CD44v6, observed with a microscope (fluorescent for the left photo and bright field for the middle one; the right photo is merged). Scale bars represent 50 µm.

**Figure 4 cancers-13-04966-f004:**
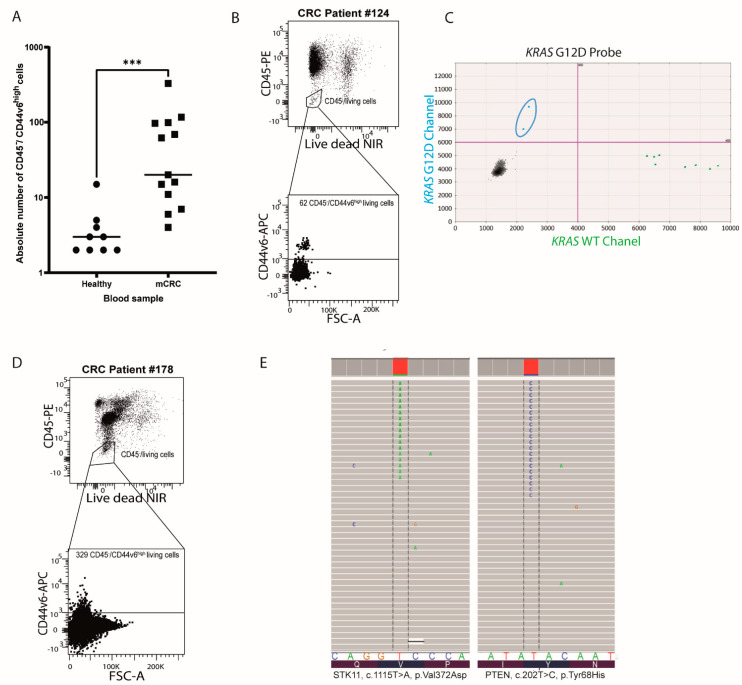
CD44v6^high^/CD45^−^CTCs can be purified from CRC patient blood samples by cytometry. (**A**) Dot plot representing a significantly higher number of CD44v6^high^/CD45^−^cells in the blood of CRC patients (*n* = 13) compared to the blood of healthy subjects (*n* = 9). The number of CD45^−^/CD44V6^high^ cells in the different groups was compared using the nonparametric unpaired Mann–Whitney U test (2-tailed, confidence interval of 95%, *** *p* value = 0.0001). (**B**) FACS plot demonstrating the presence of live CD45^−^/CD44V6^high^ cells in the blood of patient no. 124. NIR stands for “near infra-red”. (**C**) DNA was extracted from the latter cells for digital droplet PCR. Scatter plot of the digital droplet PCR results shows fluorescent detection of individual droplets. Green (x-axis) and blue (y-axis) dots represent droplets with KRAS WT and KRAS G12V (surrounded in blue) genotypes, respectively. (**D**) FACS plot demonstrating the presence of live CD45^−^/CD44V6^high^ cells in the blood of patient no. 178. (**E**) DNA was extracted from the latter cells for next-generation sequencing. (**E**) Molecular alterations detected by sequencing in the blood of patient no. 178. The STK11 V372D mutation was present at 8.7% (**left**), and the PTEN Y68H alteration was detected at 6.2% (**right**)—visualized using the Integrative Genomic Viewer program.

**Figure 5 cancers-13-04966-f005:**
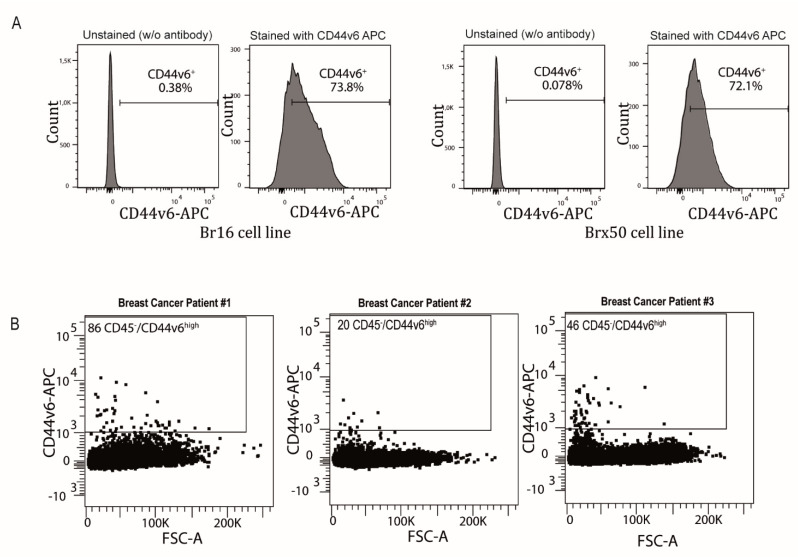
CD44v6 is also expressed in CTCs from breast cancer. (**A**) Plots demonstrating the high expression of CD44v6 in two established breast CTC lines: Br16 and Brx50. (**B**) FACS plots demonstrating the presence of CD45^−^/CD44V6^high^ cells in the blood of patients with breast cancer (*n* = 3). FSC stands for “forward scatter”.

**Table 1 cancers-13-04966-t001:** Patient Clinical Information.

Patient #	Sex	Age	Primary Tumor Localization	Identified Mutations	Method	Metastatic Sites
99	Male	61	Caecum	*Kras* G13D	ctDNA	Peritoneum
124	Male	40	Transverse colon	*Kras* G12V	ctDNA	Peritoneum and liver
129	Male	79	Right colon	*Kras* G12V	NGS on primary tumor biopsy	Liver
130	Female	43	Rectum	no	NGS on primary tumor biopsy	Liver and anal canal
134	Female	62	Right colic flexure	*Braf* V600G	NGS on primary tumor biopsy	Liver and Lung
143	Male	73	Sigmoid colon	no	NGS on primary tumor biopsy	Liver
144	Male	70	Transverse colon	*Kras* G12V	ctDNA	Liver and Lung
149	Male	62	Right colon	*Kras* G12V	NGS on primary tumor biopsy	Liver
170	Female	48	Right colon	*Kras G12D*	PCR and sequencing	Liver and bones
174	Female	59	Transverse colon	*STK11*	NGS	Peritoneum and liver
178	Female	55	Rectum	*Kras G12C*	PCR and sequencing	Liver and Lung
180	Female	69	Rectum	*Kras* G12V and PTEN	NGS	Lung
183	Male	71	Right colon	*BRAFV600E and IDH1*	NGS	Peritoneum

**Table 2 cancers-13-04966-t002:** Mutations detected in CTCs from patient number 99’s blood sample by NGS.

Patient #99							
Genes	Locus	Location	Genotype	Type	Amino Acid Change	% Frequency	Variant Effect
ERBB4	chr2: 21228869	*ERBB4*: exonic:NM_005235.2	G/A	SNV	p.Thr926Met	54.65	missense
FGFR1	chr8: 38285930	*FGFR1*: exonic:NM_001174067.1	CATC/C	INDEL	p.Asp160del	CAT = 0.00, C = 3.38	non frameshift deletion

## Data Availability

Not applicable.

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
