# Peer review of "CD44v6 Defines a New Population of Circulating Tumor Cells Not Expressing EpCAM"

_cancers, 2021, doi:10.3390/cancers13194966_

Round 1
Reviewer 1 Report
The authors submit a revised version of the manuscript showing the application of CD44v6 as a target for the isolation of CTCs in colorectal and breast cancer patients. As compared to the first version, the manuscript has slightly been improved.
Still there are some major and minor issues:
Major points:
Add reference in line 63 (e.g. DOI: 10.1002/cyto.a.23571)
Add "prostate cancer" to line 67
Add refs to line 67 (Cellsearch in the three cancer types)
Add ref to line 83
Line 93-98: were all 13 CRC Patients assessed for CD44v6 and EpCAM expression? This information could be added to the paragraph
Line 115: are these 20 CRC patient the same cohort than shown in Table 1 (plus additional 7 pat)? In other words: can the CTC numbers of some patients be compared in parallel sample processing?
Figure 2B: scale bar is missing.
line 127: arrow heads are green and blue, not black and white.
line 129: omit "(C) and (D)...bead isolation". Instead, add "in the newly established CTC line" in line 130
Figure 2C: increase size quality of axis lables.
Line 130: ref to suppl. method is missing for pyrosquencing
Line 145: add number of experiements (n=?)
Figure 3B: does the green color correspond to GFP? Showing the APC signal would make more sense.
Figure 3B: the cells are very large, are they clusters? discuss
Line 154: specify the leukocyte number after immunmagnetic separation and FACS, respectively
Line 168: ref to suppl. method is missing for ddPCR
Line 175: ref to suppl. method is missing for NGS
Figure 4A: is a dotplot, not a histogramm. Absolute or relative amount of CTCs on the y-axis?
Figure 4B and D: increase size of text inserts in the FACS images
Figure 4E: increase size of labels on x-axis
Figure 5A: The text insert in the FACS plots are the same, but should probably be with and without secondary antibody?
Line 208: check if [12] is the correct reference. Anyhow, the reference is not necessary here
Line 231: Stassi is not reference 16
Line 250: Check if refs 27 and 28 are indeed previous data from your group (27 is a chinese paper, and 28 groups from Belgium and Australia)
Line 294: How were the isolated cells after magnetic separation visualized? By APC?
Line 295ff: Restructure the paragraph on flow cytometry. I suggest to change the title to "Flow cytometry and sorting". Line 296 "The cells..." means which cells - those after red blood cell lysis? No information on the FACS device is given, nor on the procedure itself
Line 312: Dead cells cannot be excluded with DAPI.
Minor points:
line 62: cite Angle correctly as Angle plc. (UK)
In line 94 it is not necessary to cite ref 15
In line 98 yout could swap cytometry and magnetic beads, in accordance to the reference/description in the manuscript
e.g. Fig.1: you could standardize the nomenclature in the entire manusript (either "CD44v6/other marker" or "other marker/CD44v6")
Table 2 header: add patient identifier
Line 199: in the cited reference no CTC line named BR16 is published
Layout of references. in line 199, 208, 235
Line 267: Add blood volume
Line 273: change to "The Red Blood Cell.....was diluted tenfold in...."
Line 283 and 297: Biotek instead Biotech
Line 404: complete sentence
Lines 413ff: harmonize intervals between the initials (preferably without intervals between initial of first name and surname, like GB - not G B)
Author Response
Dear reviewer,
Please find the response to all the points that you have raised in your report.
I hope that these modifications will convince you
Sincerely yours

Reviewer 2 Report
Even if the authors have strategically argued some of the observation made , they have assessed issues raised in the revision making the manuscript suitable for publication.
Author Response
Dear reviewer,
Authors would like to thank you for having said that the revised form of the manuscript is now suitable for publication
Sincerely yours
Round 2
Reviewer 1 Report
The authors submitted a revised version of the manuscript considering all minor and major points. In the present version all critical points had been addressed and the maunscript was amended accordingly.
This manuscript is a resubmission of an earlier submission. The following is a list of the peer review reports and author responses from that submission.
Round 1
Reviewer 1 Report
Belthier et al. CD44v6 as a novel marker for CTC detection. On the whole, the study is somewhat convincing but in my opinion did not provide enough evidence regarding the methods and data to warrant publication yet.
Major comments:
1) Line 61 - The authors state that "Identification of transmembrane CTC markers is thus urgently needed." Why is there an urgent need to develop transmembrane CTC markers?
2) What is the benefit of using transmembrane CTC markers that highlight stem cell-like CTCs? Would you miss CK+ CTCs (which most technologies isolate and has been validated clinically) by using a transmembrane marker for only capturing stem cell-like CTCs?
3) Line 82 and Line 86 - why is this data not shown? Evidence is needed since this seemed to be an important experiment.
4) Line 96 - what was the cell death marker?
5) Line 98 - "For this strategy, the patient derived CTC45 were GFP-labelled to facilitate their tracking during the whole process". No details are given in the methods as to how these were labelled, please provide this.
6) Line 112 - for non-specialist readers, please provide details as to why contamination of CD45-positive cells could be deleterious for remaining cell survival.
7) Line 114 - what was the threshold for determining CD44v6high? Are there CD44v6low cells? Specific criteria is required for this definition.
8) For the ddPCR experiments, what controls were performed? Was matched patient germline DNA also analysed to rule out clonal hematopoiesis? There are very few droplets present in the trace shown, suggesting that very little DNA went into the reaction. The authors state that only 2 copies of mutant DNA was present in the CTCs and tumour. This may just be noise in the system. Further details are required here.
9) Line 153 - The authors place a little too much emphasis on the significance of this result. CTCs were detected in 3 patients, and the standard deviation is very large. This should be highlighted and a note of caution given.
10) According to the data provided (although i haven't got the numbers and am extrapolating from the figures), CTCs were also detected in all the healthy controls. Can the authors suggest how this can be a reliable CTC marker if they are also detecting CTCs in all control samples?
11) Line 241 - How were the cells counted?
12) Line 285 - Which Oncomine panel was used?
13) Line 297 - please provide details of the assay numbers from BioRad for the KRAS assay. If they were designed in house, please provide sequences.
Reviewer 2 Report
The authors describe the application of CD44v6 as a marker for capturing circulating tumor cells from patients with CRC. From previous studies showing the expression of this variant in cell lines and established CTC lines, they conclude that this variant may be an appropriate bait for the isolation of CTCs.
The approach is interesting and might be applicable to other cancer types as well (as is indicated in the manuscript with breast cancer); however, the manuscript requires much improvement before publication.
Major issues of criticism are:
Procedure of spiking has to be described in more detail in material and methods (e.g. how exactly 100 tumor cells are added, GFP labeling etc.)
Rationales of procedure 1 and 2 (MACS and FACS) are unclear. Relevant data (recovery rate of each individual exp.) need to be shown. Unclear why procedure 1 is applied in 19 CRC patients, and procedure 2 in none? Line 108-112 refers to procedure 2?
Negative controls are missing (e.g. CD44v6 in healthy donors, ddPCR in CD45+ cells) as well as positive controls (e.g.ddPCR in CD44v6+/CD45- cells).
Sensitivity and specificity of ddPCR in CTCs without prior amplification is questionable (also indicated by just 2 dots in Fig 3C)
Likewise FACS is questionable for the detection of CTCs in terms of sensitivity, especially in CRC with very low CTC numbers (see also line 169)
Reviewer 3 Report
Belthier et al in the manuscript titled “Identification of CD44v6 as a new circulating tumor cell marker for colorectal and breast cancers” demonstrated that CD44v6 could be useful to count and purify circulating tumor cells from different cancers.
In my opinion many controls are missing that would strengthen the data shown.
The authors need to give more description about CD44v6 expression should exhibit the expression obtained through cytometry tests with a recovery of 63.3% after elution
I believe it is necessary to indicate the number of live cells obtained from 19 colorectal cancer patients . From obtained cells the authors should specify the number of cells used to start culture and the timeframe utilised for the culture specifying the growth rate exhibited. In Fig 1B live cells are shown but it is not clear if the CD44v6 cells grow as single cells or as spheres; do the four panels indicate different culture time? How many steps do cells grow? Are they clonogenic?
Of the 19 cell samples obtained, only one of these was used, what characteristic did it have? Why weren't the other 18 used?
The authors need to give more description on the cell number utilized for the injection in the nude mice, was there an increase in tumor formation compared to inoculating the CTCs compared to bulk?
In Fig 2 should be specified the percentage of positivity that the authors intend with CD44v6 high
In Fig. 3c, the mutation profile of the patient needs to be included.
Furthermore, in Fig. 4b, the negative control and the percentage of the cd44v6 cells are absent.
The manuscript is not fluent to read, thus a revision on the exposition is required.